# DYNAMICCONTROL : ADAPTIVE CONDITION SELECTION FOR IMPROVED TEXT-TO-IMAGE GENERATION

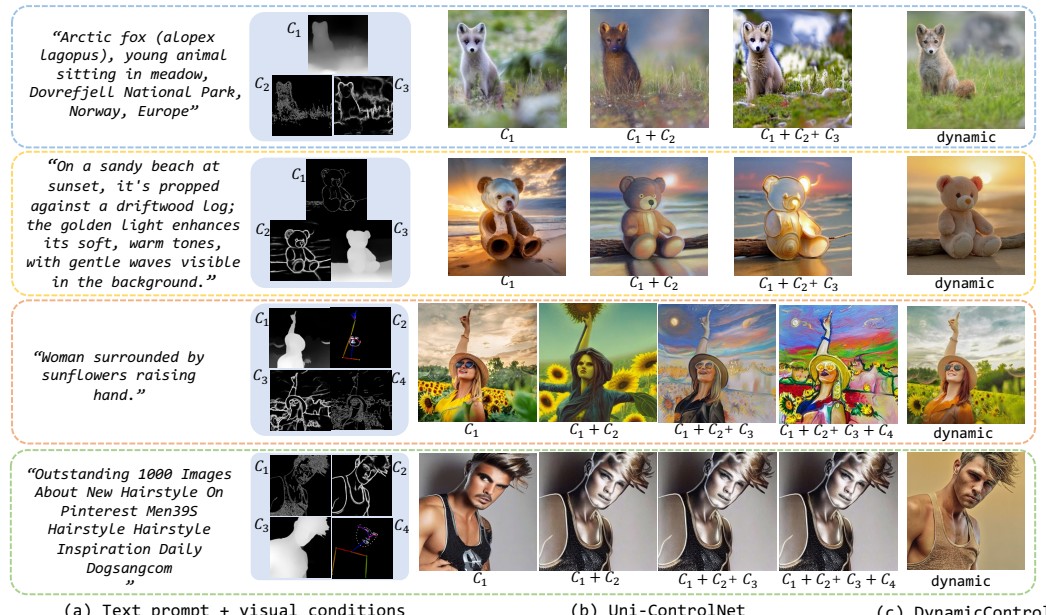

Figure 1: **Visualization comparison between method of fixed number control conditions (Uni-ControlNet (Zhao et al., 2024)) and our proposed method in different conditional controls with the same text prompt.** *(a, left two columns)* Text and various visual controls, where $C_1$, $C_2$, $C_3$ and $C_4$ denotes different control conditions. *(b, middle three or four columns)* Generation results from UniControlNet. *(c, last column)* Generation results from our DynamicControl. Previous methods struggled to generate coherent results under multiple conditions, while our results maintain strong similarity to the respective visual controls.

## ABSTRACT

To enhance the controllability of text-to-image diffusion models, current ControlNet-like models have explored various control signals to dictate image attributes. However, existing methods either handle conditions inefficiently or use a fixed number of conditions, which does not fully address the complexity of multiple conditions and their potential conflicts. This underscores the need for innovative approaches to manage multiple conditions effectively for more reliable and detailed image synthesis. To address this issue, we propose a novel framework, DynamicControl , which supports dynamic combinations of diverse control signals, allowing adaptive selection of different numbers and types of conditions. Our approach begins with a double-cycle controller that generates an initial real score sorting for all input conditions by leveraging pre-trained conditional generation models and discriminative models. This controller evaluates the similarity between extracted conditions and input conditions, as well as the pixel-level similarity with the source image. Then, we integrate a Multimodal Large Language Model (MLLM) to build an efficient condition evaluator. This evaluator optimizes the ordering of conditions based on the double-cycle controller's score ranking. Our method jointly optimizes MLLMs and diffusion models, utilizing MLLMs' reasoning capabilities to facilitate multi-condition text-to-image (T2I) tasks. The final sorted conditions are fed into a parallel multi-control adapter,

which learns feature maps from dynamic visual conditions and integrates them to modulate ControlNet, thereby enhancing control over generated images. Through both quantitative and qualitative comparisons, DynamicControl demonstrates its superiority over existing methods in terms of controllability, generation quality and composability under various conditional controls. The code and models will be available for further research.

# 1 INTRODUCTION

The emergence of generative diffusion models (Dhariwal & Nichol, 2021; Rombach et al., 2022; Ho et al., 2020; Song et al., 2020a;b) has revolutionized image synthesis tasks, attributing to the significant enhancements in the quality and variety of generated images. Built upon ControlNet-like models (Zhang et al., 2023; Li et al., 2025), various control signals such as layout constraints, segmentation maps, and depth maps have been explored to dictate the spatial arrangement, object shapes, and depth of field in generated images (Mou et al., 2024; Qin et al., 2023; Ye et al., 2023; Hu et al., 2023; Zhao et al., 2024; Sun et al., 2024). As noted in (Sun et al., 2024), different visual control signals have complementary properties. For instance, depth maps can effectively govern spatial relationships between objects but fall short in capturing fine-grained object details, while canny maps excel at capturing precise texture contours yet overlook the global structural context. In practical use cases, it is often necessary to describe the visual features of a key object through multiple visual conditions to achieve accurate control over its generation. Users generally aim to control both the overall layout and intricate details simultaneously. However, integrating visual conditions that contain rich layout and detail information into a single visual condition map remains a challenge.

Given the multiple conditions of a subject, one line (*e.g.* UniControl (Qin et al., 2023), Uni-ControlNet (Zhao et al., 2024)) chooses to activate one condition at a time during the training process randomly. This capacity to handle diverse visual conditions is quite inefficient and will greatly increase the computational burden and time costs of training. Another line of methods (*e.g.* AnyControl (Sun et al., 2024), ControlNet++ (Li et al., 2025)) uses a fixed number (usually 2 or 4) of conditions and adopts MoE design or multi-control encoder to solve the varying-number conditions problem.

However, this fixed number scheme does not fundamentally solve the problem of multiple conditions, nor does it consider whether multiple conditions conflict with the generated results. As shown in Fig. 1(b), it is suboptimal to select only one or a fixed number of conditions in previous methods without considering their importance in generating an image closer to the source image and the internal relationship between each condition. While these methods have expanded the feasibility and applications of controlled image generation, a clear and comprehensive approach to enhance controllability under diverse conditions remains an area of ongoing research and development. This highlights the need for continued innovation in integrating and optimizing control mechanisms within T2I diffusion models to achieve more reliable and detailed image synthesis.

To address this issue, we propose *DynamicControl* , a new framework that supports dynamic combinations of diverse control signals, which can adaptively select different numbers and types of conditions, facilitating more harmonious and natural generation results, as shown in Fig. 1(c).

Specifically, we begin by designing a double-cycle controller that aims to generate the initial real score sorting for all the input conditions. Within the double-cycle controller, a pre-trained conditional generation model is utilized to generate an image based on each given image condition and text prompt, then we extract the corresponding image condition from the generated image using pre-trained discriminative models. Thus, the first cycle consistency is defined as the similarity between the extracted condition and each input condition. Furthermore, considering the pixel-level similarity of source image, the second cycle consistency is performed in the calculation of the similarity between the generated image and the source image. Combining the two similarity scores, this double-cycle controller will give the combined score ranking. However, this ranking requires generating initial images for all the conditions with random noise and the source image cannot be acquired during inference, which limits its full potential. To address these limitations, we introduce the Multimodal Large Language Model (MLLM) (*e.g.*, LLaVA) (Liu et al., 2024b; Zhu et al., 2023) into our model to build an efficient condition evaluator. This evaluator takes various conditions and

promptable instructions as input and optimizes the best ordering of the conditions with the score ranking from the double-cycle controller. With a dynamic selection scheme, the final sorting results from the pre-trained condition evaluator are fed into the parallel multi-control adapter to learn necessary different level feature maps from dynamic visual conditions, where unique information from different visual conditions is captured adaptively. In this way, only those control conditions that are harmonious and mutually advantageous to the generated results are preserved. The output embeddings can be integrated to modulate ControlNet (Zhang et al., 2023), facilitating task-specific visual conditioning controls. Consequently, our DynamicControl promotes enhanced and more harmonious control over the generated images. Our main contributions are summarized as follows:

- **New Insight:** We reveal that current efforts in controllable generation still perform suboptimal performance in terms of controllability, fail to fully and effectively harness the potential of multiple conditions. And we propose the dynamic condition selection scheme, avoiding the generated images exhibit substantial deviations from the specified input conditions.

- **Efficient Condition Evaluator Learning:** We leverage MLLMs to build a condition evaluator that produces the consistency ranking score for the multiple conditions, with the supervision from an auxiliary double-cycle controller.

- **Flexible Multi-Control Adapter:** We propose a novel dynamic multi-control adapter that incorporates a series of alternating multi-control fusion and alignment blocks, designed to choose conditions adaptively and facilitate an in-depth comprehension of multi-modal user inputs.

- **Promising Results:** We provide a consolidated and public evaluation of controllability across diverse conditional controls, and illustrate that our DynamicControl comprehensively outperforms existing methods.

## 2 RELATED WORK

**Text-to-image Generation.** Text-to-Image (T2I) diffusion models (Nichol et al., 2021; Ramesh et al., 2022; Rombach et al., 2022; Saharia et al., 2022) have rapidly evolved as a leading approach for generating high-quality images from textual prompts, offering a fresh perspective on image synthesis (Kingma et al., 2021; Ho & Salimans, 2022; Dhariwal & Nichol, 2021). Initially rooted in image generation, diffusion models (Ho et al., 2020; Song et al., 2020a) have been adeptly tailored to the T2I domain, utilizing a process that incrementally introduces and then removes noise, allowing for the progressive refinement of image quality (Podell et al., 2023; Ramesh et al., 2021; Ronneberger et al., 2015; Raffel et al., 2020). This iterative denoising process, coupled with the ability to condition on both text inputs and intermediate image representations, enhances control over the generation process. Recent advancements (Gal et al., 2022; Meng et al., 2021; Brooks et al., 2023; Kawar et al., 2023; Cao et al., 2023; Huang et al.) in T2I diffusion models have incorporated various techniques to improve alignment between textual and visual features. Other models, including notable variants like DALLE-2 (Ramesh et al., 2022) and Stable Diffusion (Rombach et al., 2022), have demonstrated superior capability in capturing fine-grained structures and textures compared to earlier generative approaches. Stable Diffusion, in particular, has scaled up the latent diffusion approach with larger models and datasets, making these models accessible to the public.

**Controllable Image Synthesis.** To achieve fine-grained control over generated images, text descriptions alone often fall short in providing detailed guidance, necessitating the integration of diverse modalities for enhanced control. For instance, instance-based controllable generation methods (Wang et al., 2024b; Zhou et al., 2024) allow for location control through more free-form inputs like points, scribbles, and boxes, while structure signals like sketches and depth maps further refine the visual output.

Recent advancements have seen the development of frameworks like ControlNet (Zhang et al., 2023), ControlNet++ (Li et al., 2025) and T2I-Adapter (Mou et al., 2024), which incorporate trainable modules within T2I diffusion models to encode additional control signals into latent representations. Moreover, unified models (Huang et al., 2022; Ham et al., 2023; Hu et al., 2023; Qin et al., 2023; Sun et al., 2024) akin to ControlNet have been proposed to handle multiple control signals within a single framework, supporting multi-control image synthesis. These models typically use fixed-length input channels or a mixture of experts (MoE) design with hand-crafted weighted sum-

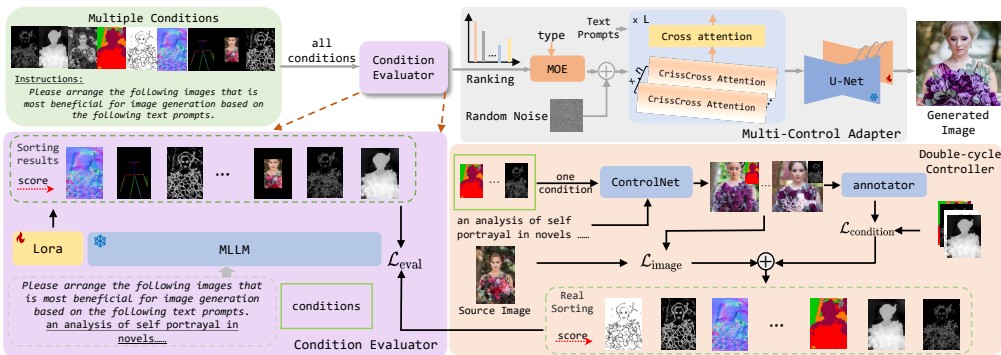

Figure 2: **Overall pipeline of the proposed DynamicControl** . For the multiple conditions, we first integrate a MLLM to build an efficient condition evaluator to rank the input conditions, which is supervised by the double-cycle controller. The ranked conditions from the pre-trained evaluator are then selected adaptively and sent into the multi-control adapter to learn dynamic visual features in parallel, thus enhancing the quality of the generated images.

mation to aggregate conditions effectively. However, despite these innovations, challenges persist in managing conditions with complex interrelations and achieving harmonious, natural results under varied control signals. Bridging the redundant information among these visual conditions and reasonably utilizing the emphasis of various visual conditions to coordinate the generation of the object is exactly the issue that this paper wants to focus on.

**MLLM with Diffusion Models.** Recent advancements in Vision Large Language Models (VLLMs) have significantly enhanced the performance of vision tasks, leveraging the extensive world knowledge and complex instruction comprehension capabilities of these models (Bai et al., 2023; Lin et al., 2023; Liu et al., 2024b;a; Chen et al., 2023). Notably, the open-sourced LLaMA model (Touvron et al., 2023) has been instrumental in improving image-text alignment through instruction-tuning, a technique further refined by models such as LLaVA (Liu et al., 2024a) and MiniGPT-4 Liu et al. (2024b); Zhu et al. (2023). These models have demonstrated robust capabilities across a variety of tasks, particularly those reliant on text generation. In the realm of image generation, fine-tuning VLLMs has shown great success (Ge et al., 2024; Li et al., 2023b; Koh et al., 2024; Ge et al., 2023a;b). For instance, SmartEdit (Huang et al., 2024b) adapts the LLaVA model to specialize in image editing tasks. FlexEdit (Wang et al., 2024a) employs a VLLM in comprehending the image content, mask, and user instructions. Additionally, models like Emu (Sun et al., 2023) and CM3Leon (Yu et al., 2023) have expanded the capabilities of multi-modal language models, employing architectures and training methods adapted from text-only models to execute both text-to-image and image-to-text generation tasks effectively.

## 3    METHOD

The pipeline of DynamicControl is demonstrated in Fig. 2. Given the multiple conditions, we first introduce the double-cycle controller (Section 3.1) to produce the real ranking score as the supervision signal for training the condition evaluator (Section 3.2) combined with MLLMs. Then, these ranked conditions with selection scores from the pre-trained condition evaluator are dynamically encoded by the multi-control adapter (Section 3.3) to fulfill controllable image generation. Finally, we discuss how to jointly optimize MLLMs with diffusion models and train our multiple conditional T2I model (Section 3.4).

### 3.1    DOUBLE-CYCLE CONTROLLER

Given that we conceptualize multi-conditional controllability as a dynamic selection among input conditions, it becomes feasible to measure this selection using a discriminative reward model. By quantifying the outputs of the generative model, we are then able to enhance the optimization of various conditional controls collectively, relying on these quantitative assessments, to facilitate more controlled generation processes.

To be more specific, given the multiple conditions along with text prompts, we first utilize a pre-trained conditional generation model (Zhang et al., 2023; Li et al., 2025) to generate images for each

condition. Then corresponding reverse conditions are extracted by different pre-trained discriminative models. Based on these generated images and reverse conditions, we design a double-cycle controller to make an initial importance assessment of the input multiple control conditions. This double-cycle controller consists of two consistency scores, namely condition consistency and image consistency.

**Condition Consistency.** Inspired by (Zhu et al., 2017; Li et al., 2025), for each input condition $c_{i,v}$ ($i = 1, 2, ..., N$, $N$ is the total number of conditions) and the corresponding output condition $\hat{c}_{i,v}$ of the generated image $x'_0$, we optimize the condition cycle consistency loss for better controllability, which is formulated as:

$$\begin{aligned} \mathcal{L}_{\text{condition}} &= \mathcal{L}(c_{i,v}, \hat{c}_{i,v}) \\ &= \mathcal{L}(c_{i,v}, \mathbb{D}[\mathbb{G}(c_{i,t}, c_{i,v}, x'_t, t)]). \end{aligned} \tag{1}$$

Here we perform single-step sampling (Ho et al., 2020) on disturbed image $x'_t$, which means $x_0 \approx x'_0 = \frac{x'_t - \sqrt{1-\alpha_t}\epsilon_\theta\left(x'_t, c_{i,v}, c_{i,t}, t\right)}{\sqrt{\alpha_t}}$, where $\mathbb{D}$ is the discriminative reward model to optimize the controllability of $\mathbb{G}$. $\mathcal{L}$ represents an abstract metric function that is adaptable to various concrete forms depending on specific visual conditions. This flexibility allows it to be tailored to meet the unique requirements of different visual analysis tasks, enhancing the applicability and effectiveness of the model across diverse scenarios.

**Reverse Image Consistency.** Apart from the condition consistency, we employ a reverse image consistency loss to guarantee that the original image is similar to the generated one. We achieve this by minimizing pixel-wise and semantic discrepancies between the generated image and the source image. Given the CLIP embeddings (Radford et al., 2021) of the source image $E_{I_{source}}$, generated image $E_{I_{gen}}$, the loss is defined as:

$$\mathcal{L}_{\text{image}} = 1 - cos(E_{I_{source}}, E_{I_{gen}}). \tag{2}$$

This loss ensures that the model can faithfully reverse conditions and return to the source image when the conditions and text instructions are applied, enforcing the model by minimizing differences between the source and generated images.

## 3.2 CONDITION EVALUATOR

Although the double-cycle controller can make a combined score ranking for the various control conditions, it remains two challenges: (i) employing a pre-trained generative model for image synthesis, regardless of its proficiency, introduces an elevated level of uncertainty in the outcomes, which means a significant reliance on the foundational generative model employed, (ii) the source image is not available during the inference, especially in user-specified tasks. To address this issue, we introduce a Multimodal Large Language Model (MLLM) into our network architecture.

As shown in Fig. 2, given the conditions $c_1, c_2, \ldots, c_N$ and instruction $\tau$, our primary objective is to optimize the best ordering of the conditions with the score ranking from the double-cycle controller. Inspired by (Koh et al., 2023; Huang et al., 2024a), we expand the original LLM vocabulary of LLaVA (Liu et al., 2024b) with $N$ new tokens "$<con^0>$, ..., $<con^N>$" to represent generation information and append these tokens to the end of instruction $\tau$. Then, the conditions $c_1, c_2, \ldots, c_N$ and the reorganized instruction $\tau'$ are fed into the Vision Large Language Model (VLLM) $LLaVA(\cdot; \omega)$ to obtain response tokens, which are processed to extract the corresponding hidden states $h_i \in \mathcal{H}$, capturing the deeper semantic information from the VLLM's representations of the inputs. However, these hidden states predominantly exist within the text vector space of the LLM, presenting compatibility issues when interfacing with a diffusion model, especially one trained on CLIP text embeddings (Radford et al., 2021). This discrepancy can hinder effective integration between the models. Considering this, we transfer Q-Former (Li et al., 2023a) to refine the hidden states into embeddings $f_c$ compatible with the diffusion model. The transformation process is represented as:

$$\begin{aligned} R &= LLaVA(c_1, c_2, ..., c_N, \tau'; \omega), \\ h_i &= H(c_1, c_2, ..., c_N, \tau'; \omega|r_i), \\ f_c &= Q(\mathcal{H}), \end{aligned} \tag{3}$$

where $r = \{$"$<con^0>$, ..., $<con^N>$"$\} \in R$ is the condition tokens set and $f_c$ represents the transformed embeddings by the Q-Former function $Q$. For fine-tuning efficiency, we utilize the

LoRA (Hu et al., 2021) scheme, where the majority of parameters $\theta$ in the LLM are kept frozen. The recurrent optimization process can be formulated as:

$$\mathcal{L}_{\text{LLM}}(\tau) = -\sum_{i=1}^{N} \log p_{\omega+\Delta\omega(\theta)}(<con^i> \mid c_1, c_2, ..., c_N, \tau'). \tag{4}$$

Subsequently, the predicted results from the LLM for each condition are supervised by corresponding ranking scores from double-cycle controller, optimizing the final sorting rankings. The process is represented as $\mathcal{L}_{\text{eval}} = -\sum_{i=1}^{N} c_i \log p_i$.

### 3.3 MULTI-CONTROL ADAPTER

To accommodate the simultaneous application of multiple dynamic control conditions, we have innovatively designed a multi-control adapter. This adapter is engineered to interpret complex control signals adaptively, enabling the extraction of comprehensive multi-control embeddings from textual prompts and dynamic spatial conditions.

After acquiring the well-pretrained condition evaluator, its robust understanding capabilities can be leveraged

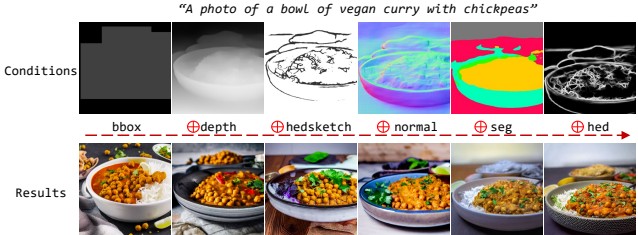

Figure 3: **Results of adding different conditions ranked by the condition evaluator.** Starting from the leftmost with the lowest score, we gradually add the control conditions with higher scores from left to right.

to score all input conditions. From the pool of scored conditions, only those that meet or exceed a predefined threshold are selected to participate in the subsequent optimization of the T2I model. This selective approach ensures that only the most relevant and high-quality conditions contribute to the training process, potentially enhancing the effectiveness and efficiency of the T2I model. Regarding the threshold setting, it is not manually predefined nor maintained consistently across all data pairs within the training set. Instead, it is configured as a parameter that is subject to learning, allowing the model to adaptively determine and adjust the threshold for various datasets. Consequently, as experimented in Sec. 4.2, this adaptive mechanism results in dynamic and diverse control conditions with no conflicts, both in quantity and type. These conditions are employed in the training process depending on the specific characteristics of each dataset. This approach ensures that the training is tailored to the unique demands and nuances of various data inputs.

As illustrated in Fig. 2, these selected conditions are then consumed by the following controllable image generation module. More details about can be found in the Appendix. Fig. 3 shows the results of the six non-conflicting control conditions selected by our adapter dynamically. The conditions with high scores are added from left to right. It can be seen that as the high-scoring control conditions are gradually added, the generated results are of higher quality and gradually closer to the text description.

### 3.4 TRAINING STRATEGY

The whole training of our network consists of two processes. The first process involves training the condition evaluator, is represented as follows:

$$\mathcal{L}_{\text{condi}} = \mathcal{L}_{\text{condition}} + \mathcal{L}_{\text{image}} + \lambda_1 \mathcal{L}_{\text{LLM}} + \lambda_2 \mathcal{L}_{\text{eval}}, \tag{5}$$

where $\lambda_1$ and $\lambda_2$ are positive constants to balance the different losses. This evaluator is then frozen, remaining unchanged during any subsequent optimization processes. The second training process involves the multi-control diffusion model.

## 4 EXPERIMENTS

We validate the effectiveness of DynamicControl on five conditions with more common control conditions: canny, hed, segmentation mask, openpose and depth. Our evaluation primarily focuses

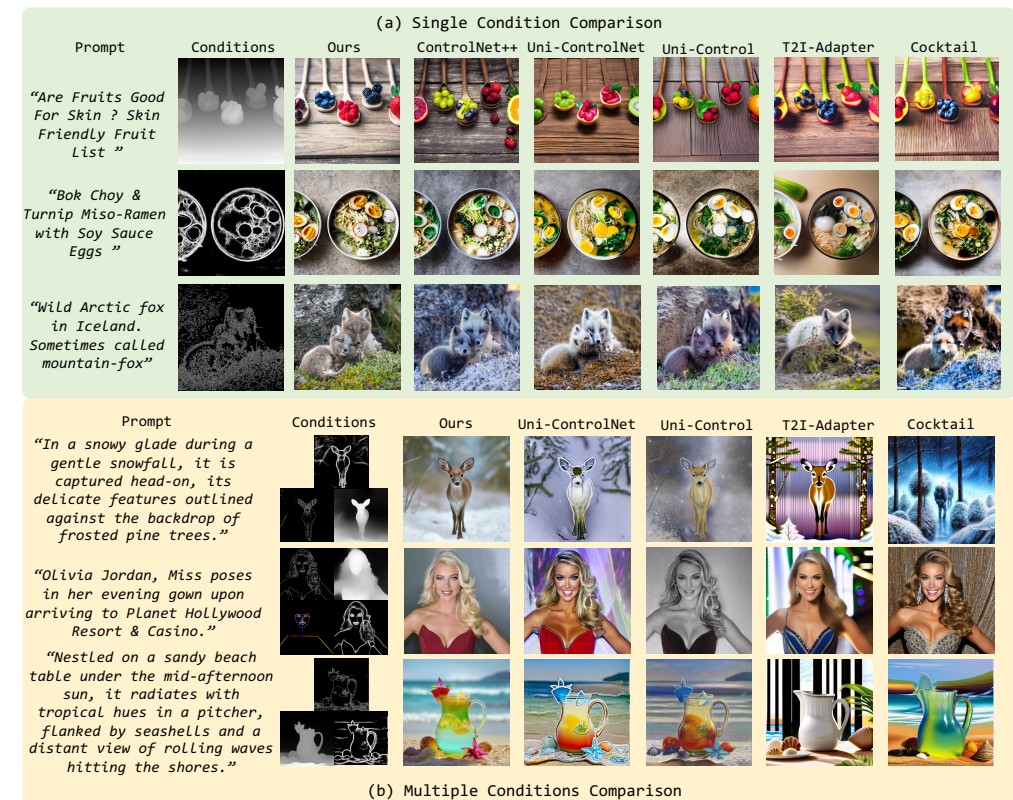

Figure 4: **Visualization comparison** between selected SOTA methods and our proposed model in different conditional controls. *Conditions* are the conditions that are ultimately used by the model.

on several leading methods in the realm of controllable text-to-image diffusion models, including Gligen (Li et al., 2023c), T2I-Adapter (Mou et al., 2024), ControlNet v1.1 (Zhang et al., 2023), GLIGEN (Li et al., 2023c), Uni-ControlNet (Zhao et al., 2024), UniControl (Qin et al., 2023), Cocktail (Hu et al., 2023) and ControlNet++ (Li et al., 2025). These methods are pioneering in their field and provide public access to their codes and model weights, which accommodate various image conditions. Although the models of other approaches such as AnyControl (Sun et al., 2024) are public, their code cannot be successfully run after many attempts. **Implementation details including network structure, datasets, evaluation metrics, computational complexities, hyperparameters of training and inference can be found in the Appendix C.**

## 4.1 MAIN RESULTS

**Comparison in Multiple Conditions.** Our DynamicControl aims to boost the control over diffusion models by multiple utilizing image-based conditions selection. As shown in Tab. 1, DynamicControl effectively addresses issues of line coarsening and subject distortion in multi - visual control conditions, as shown by quantitative (improved FID and leading MUSIQ scores) results. It also shows superior performance under full conditional control in terms of CLIP score, balancing visual control and text adherence better than other methods. Moreover, it maintains high consistency across different condition combinations and outperforms others in relevant metrics, especially under full-condition control.

**Comparison of Image Quality and CLIP Score.** To ascertain whether enhanced controllability correlates with a reduction in image quality, we present the Fréchet Inception Distance (FID) metrics across multiple conditional generation tasks, as detailed in Tab. 3. We can find that our model quantitatively reveals superior performance across all conditions compared to existing approaches. This significant achievement indicates that DynamicControl effectively handles intricate combinations of multiple spatial conditions, producing high-quality, coherent outcomes that align well with the spatial conditions. Further, to address concerns about its impact on text controllability, we employ CLIP-Score metrics to evaluate different methods across various datasets,

Table 1: **Comparison of generation quality, controllability, and text-image consistency on the MultiGen-20M (Li et al., 2025) and Subject-200K (Tan et al., 2024) datasets in multiple conditions.** "all" under the **Conditions** column represents that all conditions are used for generation.

| Methods | Conditions | MultiGen-20M | | | | Subject-200K | | | |
|---|---|---|---|---|---|---|---|---|---|
| | | FID (↓) | CLIP Score (↑) | SSIM (↑) | MUSIQ (↑) | FID (↓) | CLIP Score (↑) | SSIM (↑) | MUSIQ (↑) |
| T2I-Adapter (Mou et al., 2024) | all | 66.95 | 71.47 | 24.03 | 57.95 | 64.72 | 74.35 | 31.76 | 55.07 |
| Uni-ControlNet (Zhao et al., 2024) | all | 32.58 | 78.08 | 29.37 | 65.85 | 44.35 | 77.40 | 37.98 | 66.75 |
| UniControl (Qin et al., 2023) | all | 25.15 | 74.09 | 35.58 | 72.05 | 30.95 | 72.96 | 47.31 | 67.50 |
| Cocktail (Hu et al., 2023) | pose+hed | 24.67 | 79.87 | 24.14 | 67.13 | 51.51 | 79.38 | 29.59 | 64.16 |
| **Ours** | all | **12.01** | **81.36** | **43.85** | **74.11** | **11.51** | **81.66** | **60.22** | **69.28** |

Table 3: **FID (↓) / CLIP-score (↑) comparison under different conditional controls and datasets.** All the results are conducted on 512×512 image resolution for fair comparisons. We generate four groups of images and report the average result to reduce random errors.

| Method | Seg. Mask | | Canny | Hed | Openpose | Depth |
|---|---|---|---|---|---|---|
| | ADE20K | COCO | MultiGen-20M | MultiGen-20M | MultiGen-20M | MultiGen-20M |
| Gligen (Li et al., 2023c) | 33.02 / 31.12 | - | 18.89 / 31.77 | - | 28.65 / 31.26 | 18.36 / 31.75 |
| T2I-Adapter (Mou et al., 2024) | 39.15 / 30.65 | - | 15.96 / 31.71 | - | 26.07 / 33.54 | 22.52 / 31.46 |
| UniControlNet (Zhao et al., 2024) | 39.70 / 30.59 | - | 17.14 / 31.84 | 17.08 / 31.94 | 27.66 / 34.58 | 20.27 / 31.66 |
| UniControl (Qin et al., 2023) | 46.34 / 30.92 | - | 19.94 / 31.97 | 15.99 / 32.02 | 24.58 / 35.01 | 18.66 / 32.45 |
| ControlNet (Zhang et al., 2023) | 33.28 / 31.53 | 21.33 / 13.31 | 14.73 / 32.15 | 15.41 / 32.33 | - | 17.76 / 32.45 |
| Cocktail (Hu et al., 2023) | 31.56 / 31.77 | 19.35 / 13.68 | 12.92 / 33.16 | 14.71 / 33.07 | 22.59 / 35.78 | - |
| ControlNet++ (Li et al., 2025) | 29.49 / 31.96 | 19.29 / 13.13 | 18.23 / 31.87 | 15.01 / 32.05 | - | 16.66 / 32.09 |
| **Ours** | **25.23 / 34.38** | **16.29 / 16.21** | **10.98 / 35.75** | **11.37 / 36.07** | **20.12 / 37.25** | **11.28 / 35.49** |

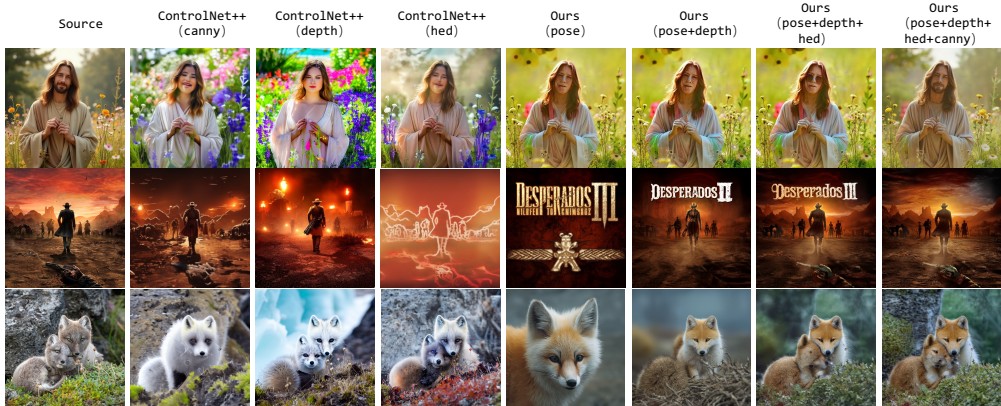

Figure 5: **Comparison of multi-condition image generation** by DynamicControl with different combinations against single-condition image generation by ControlNet++.

ensuring that the generated images closely match the input text. As shown in Tab. 3, DynamicControl achieves superior CLIP-Score results on several datasets relative to existing methods. This indicates that our approach not only significantly improves conditional controllability, but also maintains the original model's capability to generate images from text.

**Condition Types Comparison.** To examine the effect of the number of condition types, Fig.5 shows the visual results under various condition combinations, with qualitative evaluation metrics provided in Tab.2. Examples in Fig.5 demonstrate that as the number of conditions increases, the layout and texture within images become increasingly refined and accurate, demonstrating that the control effects of different visual conditions are not identical. Furthermore, as can be seen from Tab.2, an increase in visual conditions is beneficial for the overall quality and controllability of the images, which aligns with the observed visual effects. As evidenced by the comparison with ControlNet++ (Li et al., 2025), integrating multi-conditions improves control precision and enables finer-grained manipulation of the generated output.

Table 2: **Results of combining different condition types.** A quantitative comparison with models controlled by single visual conditions and various combinations of visual conditions. "Source" refers to the original reference image.

| Methods | Conditions | MultiGen-20M | | |
|---|---|---|---|---|
| | | FID (↓) | SSIM (↑) | MUSIQ (↑) |
| Source | - | - | - | 69.30 |
| ControlNet++ | canny | 17.69 | 36.69 | 65.67 |
| ControlNet++ | hed | 13.93 | 42.12 | 71.22 |
| ControlNet++ | depth | 17.56 | 27.79 | 71.23 |
| Ours | pose | 29.68 | 27.88 | 58.62 |
| Ours | pose+depth | 19.85 | 31.69 | 61.02 |
| Ours | pose+depth+hed | 13.55 | 41.52 | 66.35 |
| Ours | all | 12.01 | 43.85 | 74.11 |

Table 4: **Ablation on loss functions.** CLIP score and FID are reported on ADE20K and MultiGen-20M datasets. *Base* is a custom reduced version of DynamicControl.

Figure 6: **Ablation on different selection schemes.** (CLIP score, SSIM) is reported on MultiGen-20M dataset.

| Loss | ADE20K | | MultiGen-20M | |
|---|---|---|---|---|
| | CLIP Score ($\uparrow$) | FID ($\downarrow$) | CLIP Score ($\uparrow$) | FID ($\downarrow$) |
| *Base* | 28.18 | 38.76 | 31.26 | 22.58 |
| $+\mathcal{L}_{condition}$ | 30.11 | 33.59 | 33.05 | 19.62 |
| $+\mathcal{L}_{image}$ | 31.23 | 29.66 | 34.22 | 18.22 |
| $+\mathcal{L}_{LLM}$ | 33.04 | 27.58 | 35.26 | 16.85 |
| $+\mathcal{L}_{eval}$ | 34.38 | 25.23 | 36.68 | 15.28 |

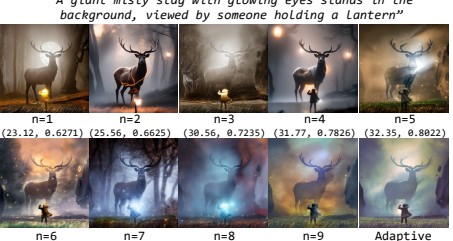

**Qualitative Comparison** In Fig. 4, we visually compare different tasks in both single and multiple conditions. Our method consistently outperforms other models in terms of both visual quality and alignment with the specified conditions or prompts. This demonstrates the effectiveness of our approach in handling a diverse range of image generation tasks, while maintaining high fidelity to the input conditions. The comparative visual analysis highlights the robustness and adaptability of our method, demonstrating its efficacy across a broad spectrum of tasks.

## 4.2 ABLATION STUDY

**Loss Functions.** To assess the effectiveness of different loss functions, we start with the base model which only contains the diffusion training loss. As shown in Tab. 4, adding the loss from the double-cycle controller significantly improves the alignment between the generated images and instruction conditions, as evidenced by the improved scores. Finally, the incorporation of combining the losses from LLM yields the best results across all metrics, underscoring the significance of condition evaluator for achieving high-fidelity and semantically accurate multiple condition generation.

**Selection Schemes.** As mentioned in Sec. 3.3, one of our key designs is to select dynamic number of conditions when performing the multi-control adapter, where we have tried different strategies of fixed numbers and adaptive number iteration. As shown in Fig. 6, in the scheme with a fixed number of iterations, the results vary with the iteration, but are still lower than those in the adaptive iteration method. This largely illustrates the effectiveness of the dynamic condition selection method in the multiple condition generation task.

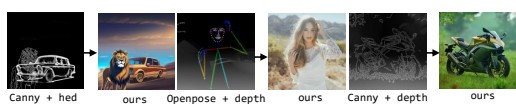

Figure 7: **Results of multiple conditions correspond to different subjects.**

**Limitation Discussion.** As shown in Fig. 7, we illustrate the results of multiple mismatched conditions, which demonstrates that our method has learned the combination of different conditions. But we still want to clarify that this non-aligned control generation is completely different from our setting. What we want to address is the complexity of multiple conditions and their potential conflicts for the same subject as we describe in the abstract and introduction. And we will dive into the exploration of this non-aligned control generation task in the future.

## 5 CONCLUSION

In this paper, we demonstrate from both quantitative and qualitative perspectives that existing works focusing on controllable generation still fail to fully harness the potential of multiple control conditions, leading to inconsistency between generated images and input conditions. To address this issue, we introduce DynamicControl , it explicitly optimizes the consistency between multiple input conditions and generated images using an efficient condition evaluator to rank the conditions, which integrates MLLM's reasoning capabilities into the T2I generation task. Experimental results from various conditional controls reveal that DynamicControl substantially enhances controllability, without sacrificing image quality or image-text alignment. This provides fresh perspectives on controllable visual generation.

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

# APPENDIX

## A    REPRODUCIBILITY STATEMENT

We have already elaborated on all the models or algorithms proposed, experimental configurations, and benchmarks used in the experiments in the main body or appendix of this paper. Furthermore, we declare that the entire code used in this work will be released after acceptance.

## B    THE USE OF LARGE LANGUAGE MODELS

We use large language models solely for polishing our writing, and we have conducted a careful check, taking full responsibility for all content in this work.

## C    IMPLEMENTATION DETAILS

**Datasets.** In our framework, we include up to 12 control conditions, the majority of which are sourced from the MultiGen-20M dataset, as proposed by UniControl (Qin et al., 2023). This dataset is a specialized subset derived from the larger LAION-Aesthetics (Schuhmann et al., 2022). More specifically, for the segmentation mask condition, our framework utilizes the ADE20K (Zhou et al., 2017; 2019) and COCOStuff (Caesar et al., 2018) datasets, following ControlNet (Zhang et al., 2023). For instances where the text caption data is missing in ADE20K, we supplement it with data from ControlNet++ (Li et al., 2025). In the testing dataset, due to the small size of the MultiGen-20M test set, and to ensure fairness in comparison, we combine the test and validation sets of MultiGen-20M to form a test set of up to 5500 text-image pairs. Additionally, to increase the confidence in our conclusions, we randomly sample 5000 text-image pairs from the Subject-200K(Tan et al., 2024) dataset as a second test set.

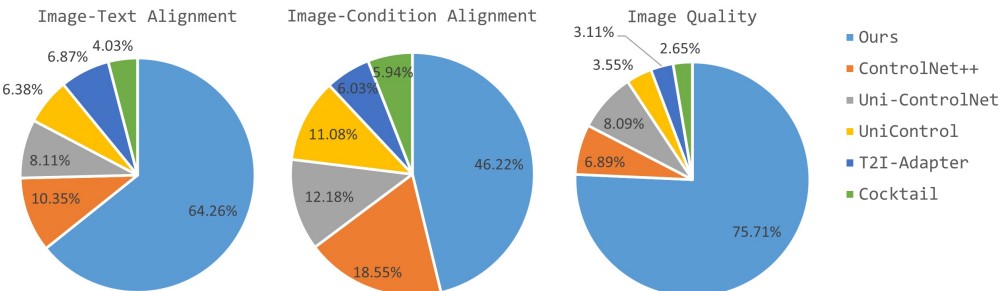

Figure A1: **The results of user studies**, comparing the results generated by ControlNet++, Uni-ControlNet, Uni-Control, T2I-Adapter and Cocktail. Based on the results from the Image-Text Alignment, Image-Condition Alignment and Image Quality perspectives, DynamicControl demonstrates superior effectiveness.

**Evaluation Metrics.** Our DynamicControl is trained using the training subsets of the respective datasets, and evaluations of all methods are conducted on the validation subsets. To ensure a fair comparison, the resolution for both training and inference in our framework is set at $512 \times 512$ for all datasets and methods involved. Since the existing methods do not have so many control conditions, in order to facilitate fair comparison, we compare the quantitative evaluation on five conditions with more common control conditions: canny, hed, segmentation mask, openpose and depth. For each condition, controllability is assessed by quantifying the resemblance between the input conditions and the conditions extracted from the images generated by diffusion models. For the evaluation of segmentation, openpose and depth controls, we employ the mIoU, mAP and RMSE as metrics respectively, which is a common practice in related research fields. In the task of canny detection, we utilize the F1-Score as it effectively addresses the binary classification of pixels into categories of 0 (non-edge) and 1 (edge). This metric is particularly suitable given the pronounced long-tail distribution observed in edge data (Xie & Tu, 2015). For evaluating other methods, we use the open-source code provided by the respective developers to generate images. We ensure a fair comparison by conducting evaluations under identical conditions, using the same datasets and without altering their inference configurations.

**Details of Multi-control Adapter.** As illustrated in Fig. 2, these selected conditions are then consumed by Mixture-of-Experts (MoEs) (Shazeer et al., 2017), where all conditions are captured in parallel as features of various low-level visual conditions. Subsequently, the extracted features are channeled into $L$ blocks, each comprising criss-cross attention (Huang et al., 2019) and cross attention mechanisms. Within each block, the input features are initially processed through $n(n \leq N)$ criss-cross attentions operating in parallel, aligning them across various feature space dimensions. Following this alignment, the diversified features are concatenated, which are then forwarded to the cross attention module. This sequential processing ensures a comprehensive integration of feature dimensions, enhancing the depth and relevance of the attention mechanisms applied. Finally, these multi-control embeddings are utilized to direct the generative process, ensuring that the output is aligned with the nuanced requirements specified by the multiple control conditions.

**Baselines.** Our evaluation primarily focuses on several leading methods in the realm of controllable text-to-image diffusion models, including Gligen (Li et al., 2023c), T2I-Adapter (Mou et al., 2024), ControlNet v1.1 (Zhang et al., 2023), GLIGEN (Li et al., 2023c), Uni-ControlNet (Zhao et al., 2024), UniControl (Qin et al., 2023), Cocktail (Hu et al., 2023) and ControlNet++ (Li et al., 2025). These methods are pioneering in their field and provide public access to their codes and model weights, which accommodate various image conditions. Other approaches such as AnyControl (Sun et al., 2024), although their models are public, their code cannot be successfully run after many attempts.

**Training Details.** During the first stage of training, we adopt the pre-trained LLaVAv1.1-7B (Liu et al., 2024b) and QFormer (Li et al., 2023a) and employ DeepSpeed (Aminabadi et al., 2022) Zero-2 to perform LoRA (Hu et al., 2021) fine-tuning. The Stable Diffusion-1.5 (Rombach et al., 2022) is diffusion model pre-trained weights. The learning rate and weight decay parameters are set to 2e-4 and 0, respectively. In the second stage, the values of learning rate, weight decay, and warm-up ratio are set to 1e-5, 0, and 0.001, respectively. We train the model for totally 50K iterations. Furthermore, we take the AdamW (Loshchilov, 2017) as the optimizer based on PyTorch Lightning (Paszke et al., 2019) for both stages. Our full-version DynamicControl is trained on 8 H20 GPUs with the batch size of 4 and the whole training process can be completed within 3 days. It takes an average of 34s to generate one image on one H20 GPU. The values of $\lambda$ in Eq.10 are experimentally tested through a series of experiments, which are set to 2 and 1.5 respectively in our practice. These values vary from 0.1 to 3 at every 0.1 interval. In all the experiments, we adopt DDIM (Song et al., 2020a) sampler with 50 timesteps for all the compared methods.

## D   MORE EXPERIMENTS

### D.1   COMPARISON OF CONTROLLABILITY

As shown in Tab. A1, we report the controllability comparison results across different conditions and datasets. Our DynamicControl significantly outperforms existing works in terms of controllability across various conditional controls. Specifically, DynamicControl obtains 4.92% and 3.22% improvements in terms of mIoU for images generated under the condition of segmentation masks. For the canny and depth conditions, DynamicControl still outperforms other methods by 2.26% on F1 Score and 5.11% on RMSE. Furthermore, apart from using SD 1.5 (Rombach et al., 2022), we also report the results of SDXL-based (Podell et al., 2023) ControlNet and T2I-Adapter. As illustrated in the table, although the SDXL-based ControlNet and T2I-Adapter exhibit improved controllability on certain specific tasks where the robustness of the text-to-image backbone does not influence its controllability for controllable diffusion models, the enhancement is modest and they are not significantly superior to their counterparts.

### D.2   COMPARISON ON COMPUTATIONAL COMPLEXITIES

As shown in Tab. A2, we report the GPU memory and inference time for one image comparison results of different methods. It is worth noting that compared with other methods, our method is still in the normal range in terms of resource consumption and inference time although we use MLLM. Other methods such as UniControl and Uni-ControlNet activate one control condition at a time. This serialized multi-condition processing will cause the inference time to increase exponentially as the number of conditions increases. In our method, we use the trained MLLM to score the input conditions during the inference phase, and then send the sorted multiple conditions to the Multi-Control Adapter for final image generation. There is no such serial sequence operation.

| Condition (Metric) Dataset | T2I Model | Canny (F1 Score ↑) MultiGen-20M | Hed (SSIM ↑) MultiGen-20M | Openpose (mAP ↑) MultiGen-20M | Depth (RMSE ↓) MultiGen-20M | Seg. Mask (mIoU ↑) ADE20K | COCO-Stuff |
|---|---|---|---|---|---|---|---|
| ControlNet (Zhang et al., 2023) | SDXL | - | - | - | 40.01 | - | - |
| T2I-Adapter (Mou et al., 2024) | SDXL | 28.03 | - | 63.89 | 39.76 | - | - |
| T2I-Adapter (Mou et al., 2024) | SD1.5 | 23.66 | - | 60.17 | 48.40 | 12.60 | - |
| Gligen (Li et al., 2023c) | SD1.4 | 26.92 | 0.5641 | 69.88 | 38.82 | 23.77 | - |
| Uni-ControlNet (Zhao et al., 2024) | SD1.5 | 27.31 | 0.6912 | 72.71 | 40.66 | 19.39 | - |
| UniControl (Qin et al., 2023) | SD1.5 | 30.83 | 0.7967 | 75.87 | 39.17 | 25.45 | - |
| ControlNet (Zhang et al., 2023) | SD1.5 | 34.66 | 0.7622 | - | - | 32.56 | 27.47 |
| Cocktail (Hu et al., 2023) | SD1.5 | 35.22 | 0.8152 | 78.82 | 35.90 | 36.55 | 29.68 |
| ControlNet++ (Li et al., 2025) | SD1.5 | 37.04 | 0.8097 | - | 28.32 | 43.64 | 34.56 |
| **Ours** | SD1.5 | **39.26** | **0.8376** | **82.63** | **23.21** | **48.56** | **37.78** |

Table A1: **Controllability comparison under different conditional controls and datasets.** We generate four groups of images and report the average result to reduce random errors.

| Ours | ControlNet++ | Uni-ControlNet | Uni-Control | T2I-Adapter | Cocktail |
|---|---|---|---|---|---|
| 40G / 34s | 48G / 30s | 32G / 33s | 40G / 39s | 32G / 43s | 80G / 55s |

Table A2: GPU memory(G) / inference time(s) comparison of different methods.

## D.3 MORE MODEL OPTIONS

Considering the model size and inference speed, we trained a lightweight model, as shown in the following table. The lightweight version further accelerates the inference speed at the expense of certain indicators. We will make these models public in subsequent open source versions for users to choose from, so as to balance the indicators and speed.

| | FID ($\downarrow$) | CLIP Score ($\uparrow$) | SSIM ($\uparrow$) | MUSIQ ($\uparrow$) | inference time ($\downarrow$) |
|---|---|---|---|---|---|
| DynamicControl-3B | 14.23 | 80.62 | 41.22 | 72.01 | 28s |
| DynamicControl-7B | 12.01 | 81.36 | 43.85 | 74.11 | 34s |

Table A3: More model options.

## D.4 USER STUDY

To further verify the effectiveness of DynamicControl , we perform a user study. Specifically, we randomly select 50 images corresponding to five different control conditions, with 10 images allocated to each condition. For each image, we obtain the results of ControlNet++, Uni-ControlNet, Uni-Control, T2I-Adapter and Cocktail, and randomly shuffle the order of these method results. For each set of images, we ask participants to independently select the three best pictures. The first one is the best picture corresponding to the text prompt (i.e., Image-Text Alignment), and the second one is the picture corresponding to the text prompt (i.e., Image-Condition Alignment) while the third one is the picture with the highest visual quality under the condition (i.e., Image Quality). A total of 30 people participate in the user study. The result is shown in Fig. A1. Notably, we can find that over 64% and 46% of participants think that the effect of DynamicControl corresponds better with the text and control conditions and more than 75% of participants prefer the results generated by our DynamicControl . These results further indicates the superiority of our DynamicControl .

## D.5 MORE ABLATION STUDY

Since these training losses fully represent the modules we proposed, we mainly perform their related ablations in the main paper. For other parts, we supplemented the relevant ablation experiments as shown in the following table. It can be seen that the removal or replacement of these parts does not affect our main experimental conclusions.

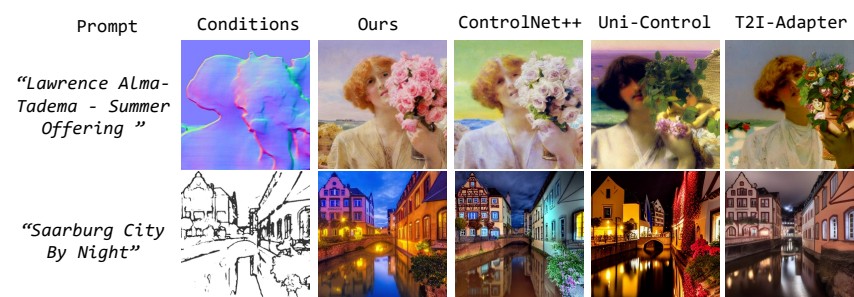

Figure A2: **Comparison on Normal and Hedsketch conditions.**

|  | FID (↓) | CLIP Score (↑) | SSIM (↑) | MUSIQ (↑) |
|---|---|---|---|---|
| base | 12.01 | 81.36 | 43.85 | 74.11 |
| w/o MoE | 12.02 | 81.35 | 43.85 | 74.12 |
| w/o criss-attention | 12.00 | 81.35 | 43.86 | 74.11 |

Table A4: GPU memory(G) / inference time(s) comparison of different methods.

## E    MORE QUALITATIVE RESULTS

More qualitative results of different conditional controls including canny, depth map, hed, human pose and segmentation map are shown in Fig. A2, Fig. A3, Fig. A4, Fig. A5, Fig. A6, Fig. A7 and Fig. A8 respectively.

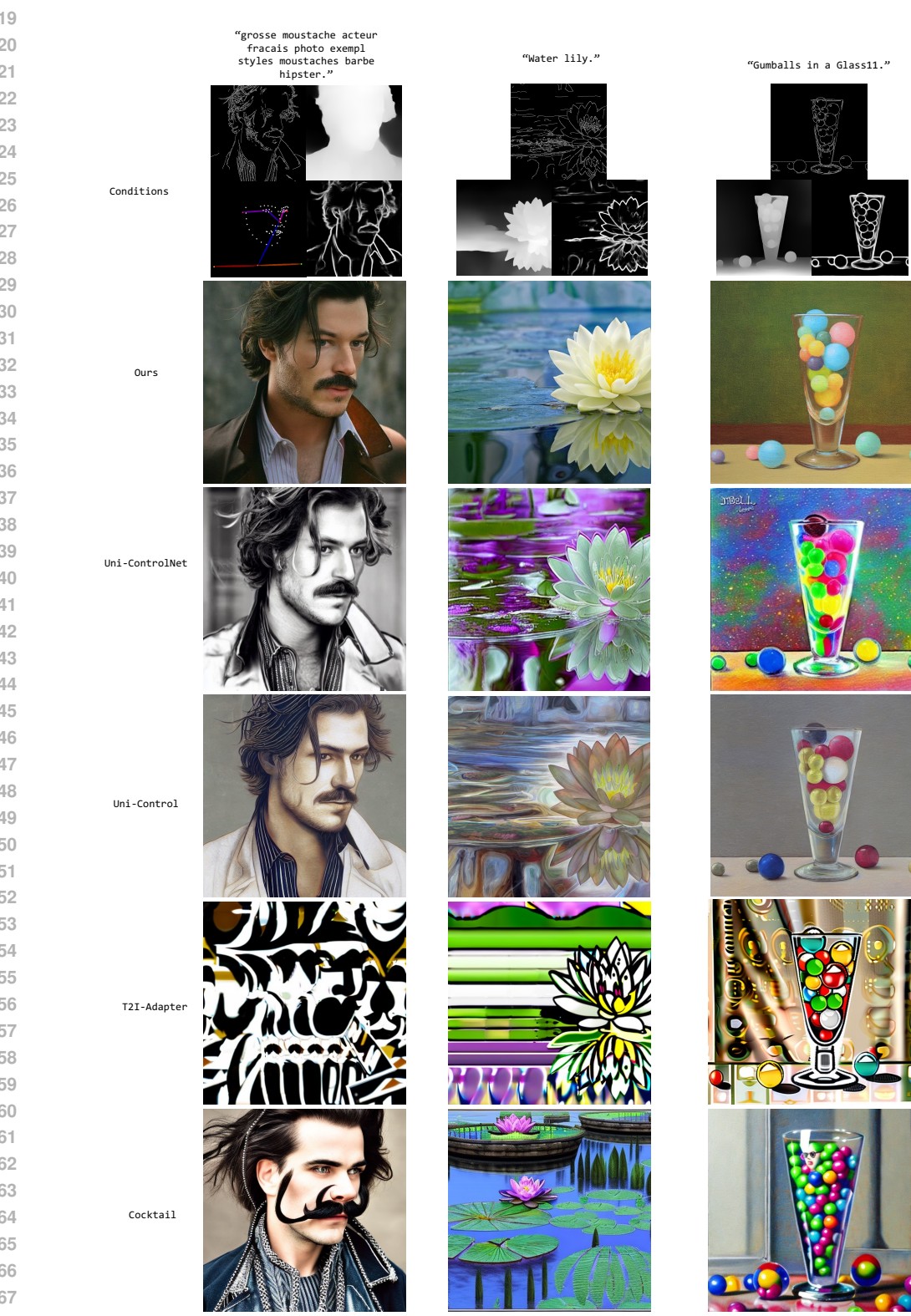

Figure A3: **Visualization comparison** between official or re-implemented methods and our proposed model in canny and depth controls with the same text prompt.

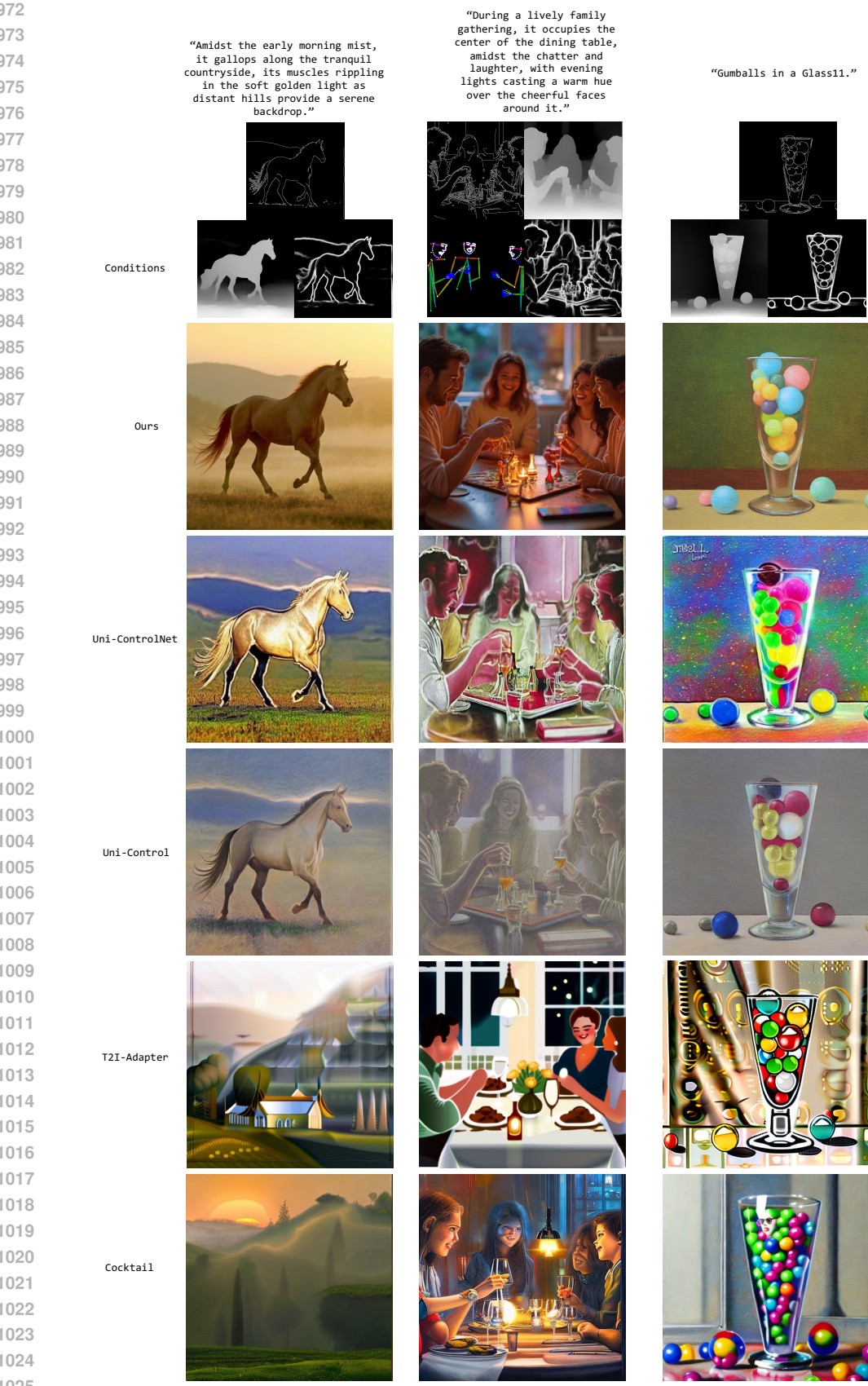

Figure A4: **Visualization comparison** between official or re-implemented methods and our proposed model in canny and depth controls with the same text prompt.

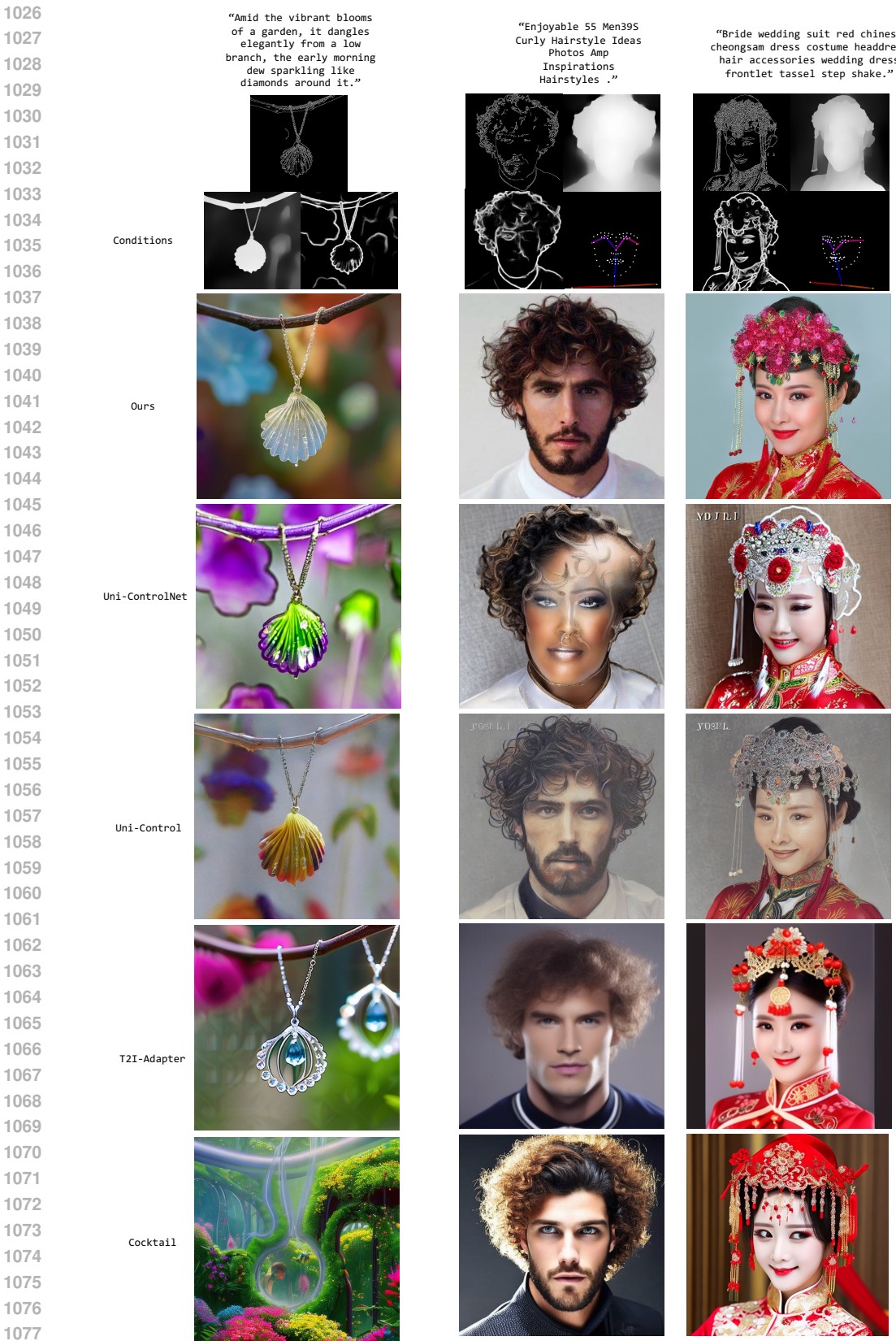

Figure A5: **Visualization comparison** between official or re-implemented methods and our proposed model in canny and depth controls with the same text prompt.

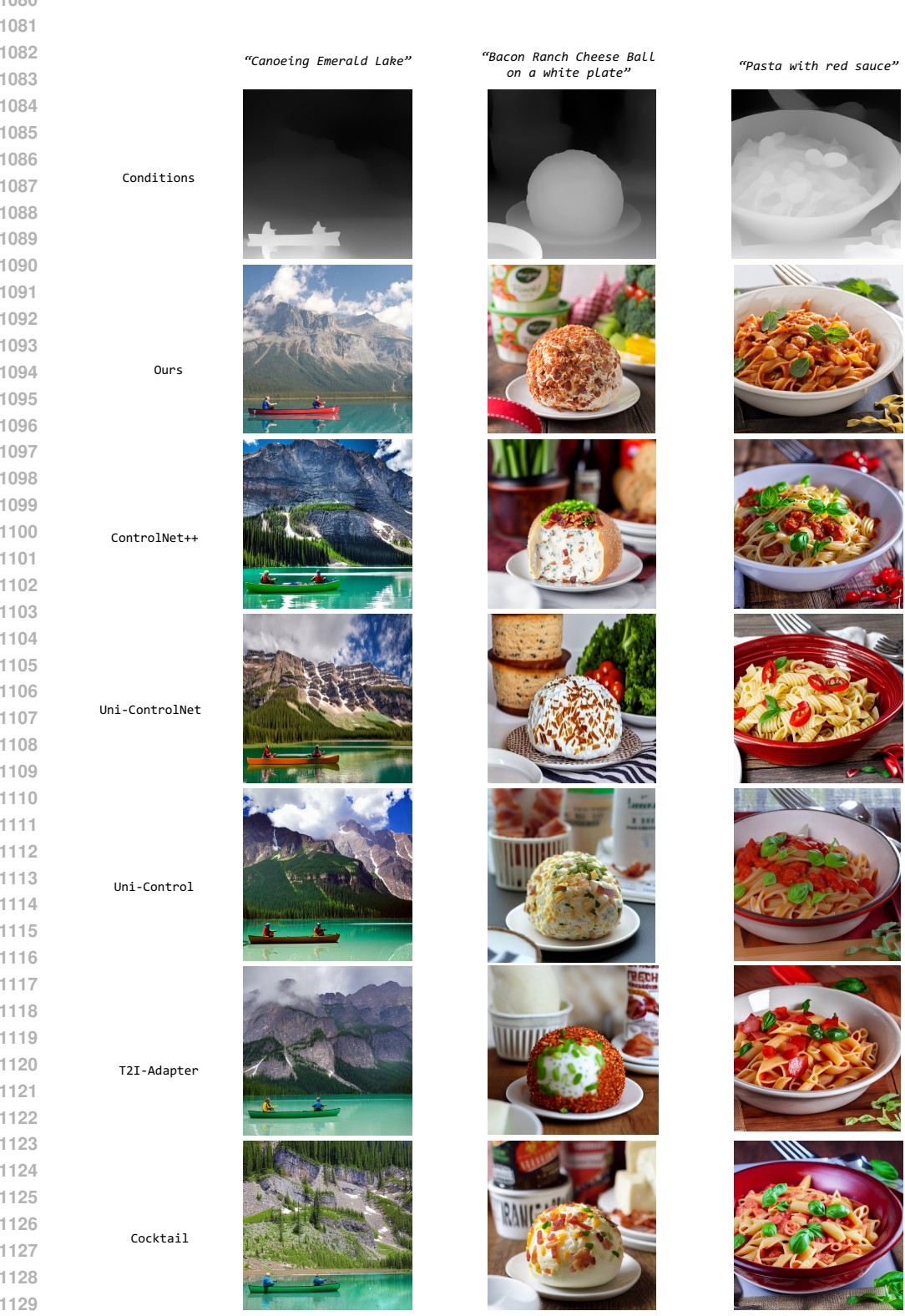

Figure A6: **Visualization comparison** between official or re-implemented methods and our proposed model in canny and depth controls with the same text prompt.

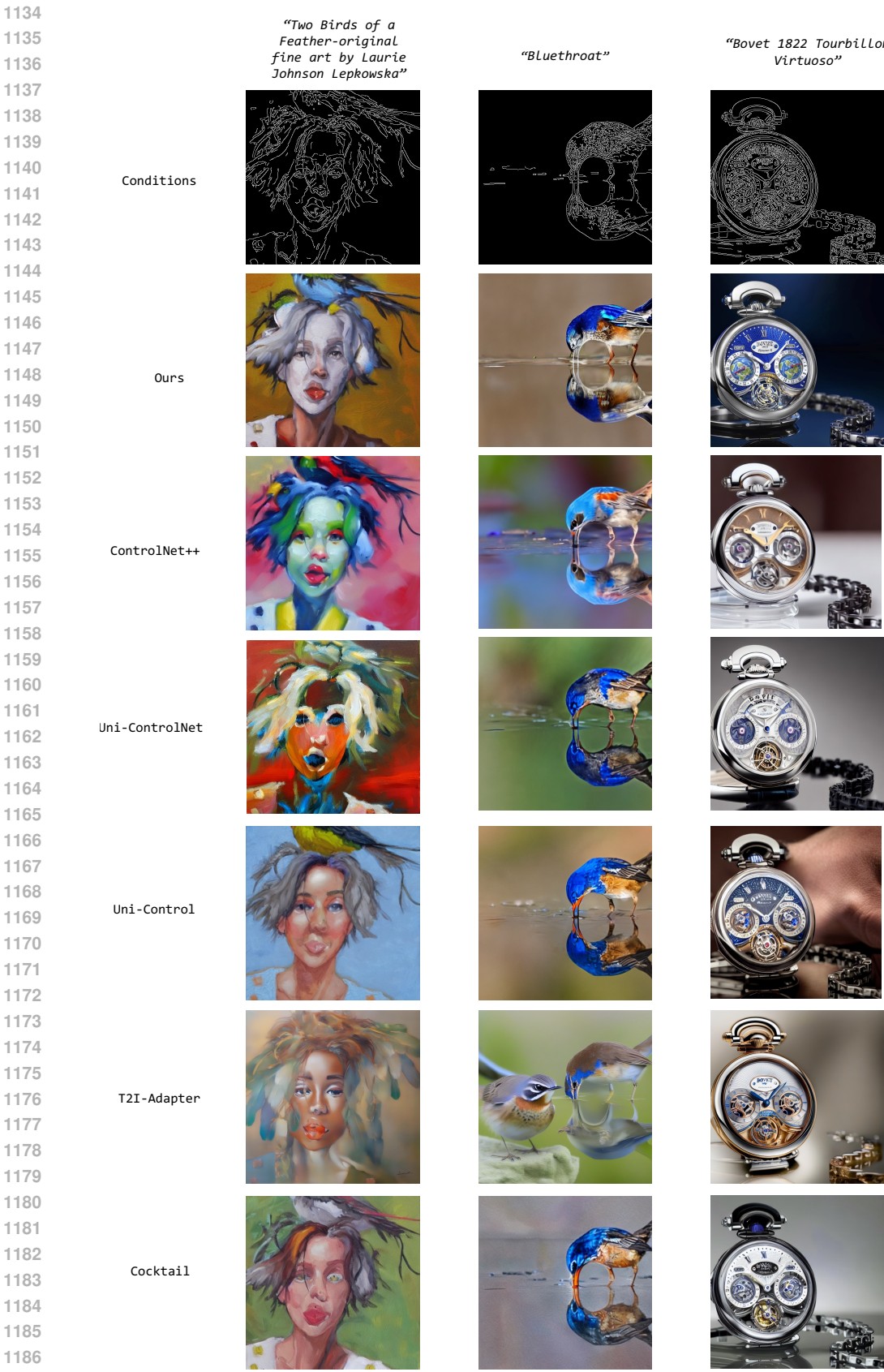

Figure A7: **Visualization comparison** between official or re-implemented methods and our proposed model in hed, openpose and segmentation maps controls with the same text prompt.

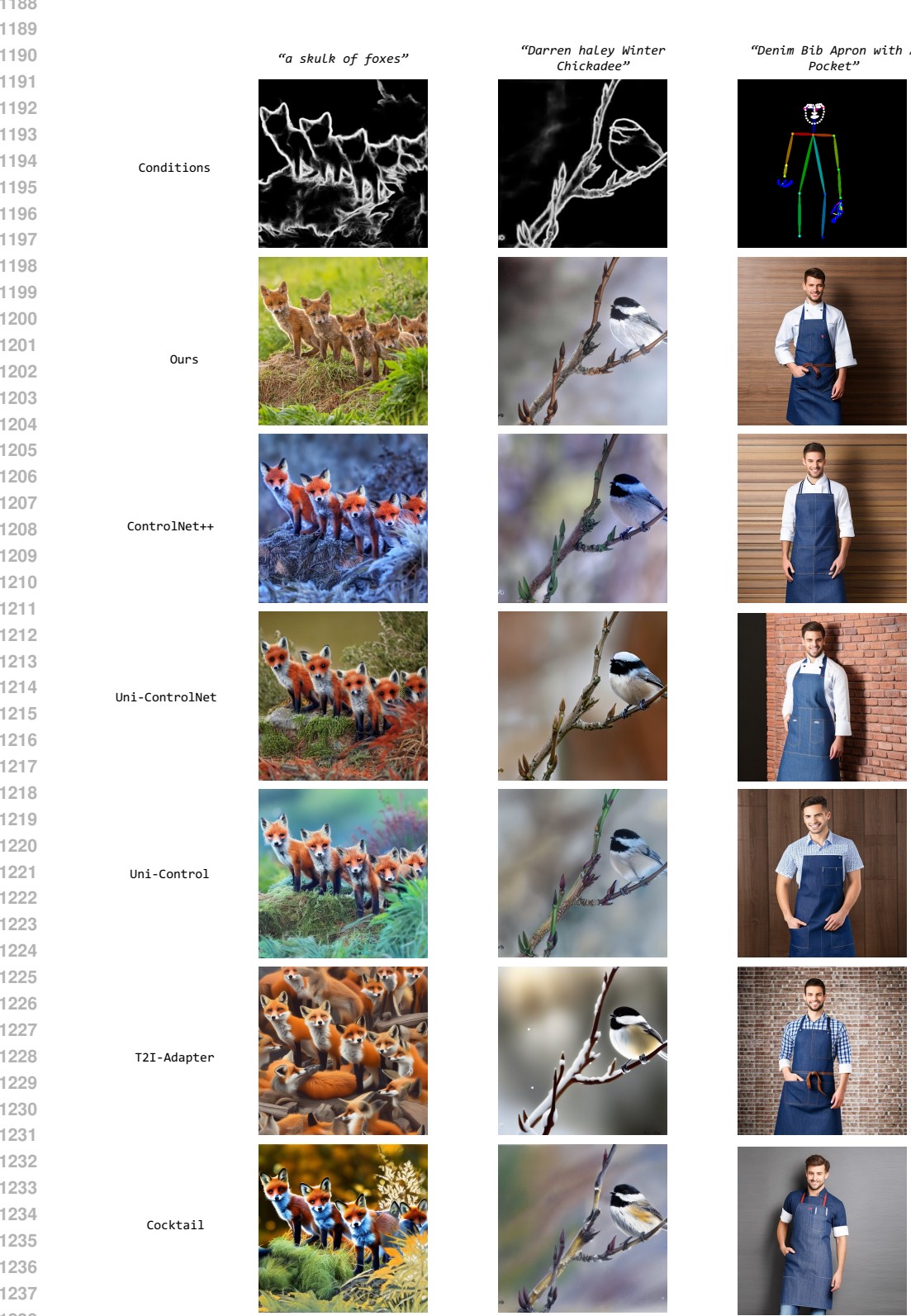

Figure A8: **Visualization comparison** between official or re-implemented methods and our proposed model in hed, openpose and segmentation maps controls with the same text prompt.

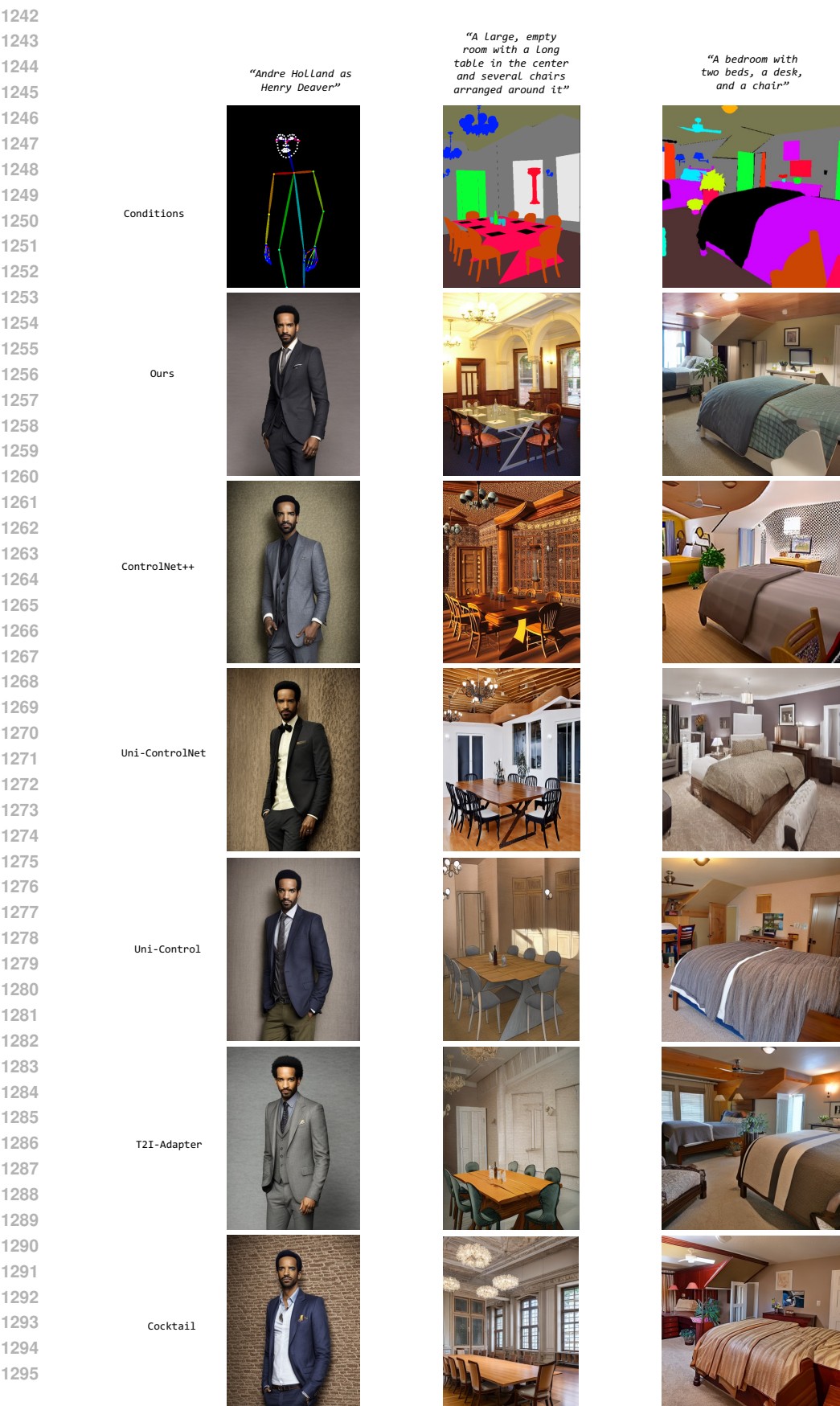

Figure A9: **Visualization comparison** between official or re-implemented methods and our proposed model in hed, openpose and segmentation maps controls with the same text prompt.

