# OpenReview forum: "DynamicControl: Adaptive Condition Selection for Improved Text-to-Image Generation"
_ICLR.cc/2026/Conference — ICLR 2026 Conference Withdrawn Submission_

### Official Review · Reviewer_RU8j · 2025-10-26

**Soundness:** 2
**Presentation:** 2
**Contribution:** 2
**Rating:** 2
**Confidence:** 4

**Summary:**

This paper proposes DynamicControl, a framework designed to address the problem of image generation under multiple control conditions. The method first computes double-cycle consistency to score different control conditions. Then, an MLLM is fine-tuned to learn the ranking of these conditions. Finally, only the conditions with scores exceeding a learned threshold are retained and passed to the Multi-Control Adapter for fusion and image generation. Both quantitative and qualitative experiments demonstrate that DynamicControl outperforms existing methods.

**Strengths:**

The qualitative and quantitative results on controllability, image quality, and image–text alignment are promising compared to competing methods, particularly in multi-control scenarios.

**Weaknesses:**

1. **Problem significance**: This paper focuses on improving controllable generation using multiple conditions derived from the **same** source image, which is not a practical problem. In real-world applications, users are more interested in handling multiple **complementary** conditions, such as combining a spatial condition with a style condition, or two spatial conditions that describe content in different regions of an image.

2. **Method rationality**: Some designs about the double-cycle controller require further clarification. For example, how metrics for different conditions are compared and why image consistency is incorporated as part of the evaluation metric. Please see “Questions” section below for detailed questions.

3. **Writing clarity**: The description of how the MLLM is trained is unclear. Certain parts, such as the formulation of the MLLM and the definition of the loss terms are confusing. Please see “Questions” section below for detailed questions.

4. **Experimental insufficiency**: As the primary contribution of this paper is introducing a framework that dynamically select conditions, the paper lacks an ablation study comparing performance when all conditions are used without dynamic selection. Such an experiment is essential to prove that the proposed dynamic selection scheme is indeed beneficial.

**Questions:**

1. For condition consistency (Line 222-233), the authors mention that different conditions use different metric functions, which implies that the resulting losses are not directly comparable (for example, SSIM scores for HED maps cannot be directly compared to RMSE scores for depth maps). Could the authors clarify how different conditions are ranked given these non-comparable losses?

2. In Line 228, why is single-step sampling used instead of a full sampling trajectory to compute $\mathcal L_\text{condition}$? Furthermore, which $t$ is used for single-step sampling, and how does the choice of $t$ affect the condition ranking?

3. In Line 234-243, why is image consistency $\mathcal L_\text{image}$ included as a metric for scoring conditions? For conditions that impose loose constraints (e.g., bounding boxes), it is acceptable for generated images to differ from the source image as long as they satisfy the condition. Using image consistency may bias the ranking toward stricter conditions.

4. In Line 262-269, how are the embeddings $f_c$ used after being transformed by the Q-Former? This notation does not appear in subsequent parts of the paper, and clarification of its role would be helpful.

5. In Line 270-279, what is the difference between $\mathcal L_\text{LLM}$ and $\mathcal L_\text{eval}$? What is the ground-truth label for $\mathcal L_\text{LLM}$, and what do $c_i $ and $p_i$ represent in $\mathcal L_\text{eval}$?

6. In Equation (5), why do $\mathcal L_\text{condition}$ and $\mathcal L_\text{image}$ appear in the loss function for training the condition evaluator? My understanding is that they serve as the targets for the MLLM rather than loss terms.

7. After the MLLM is fully trained, is the condition ranking stable (i.e., some conditions consistently rank higher than others) or does it vary across samples? Some analysis on the behavior of the trained model would be highly insightful.

---

### Official Review · Reviewer_uEv9 · 2025-10-31

**Soundness:** 3
**Presentation:** 3
**Contribution:** 1
**Rating:** 2
**Confidence:** 4

**Summary:**

DynamicControl is a framework aimed at enhancing controllability in text-to-image diffusion models by efficiently handling multiple and diverse control signals. Unlike previous models that either rely on a fixed number of conditions, DynamicControl supports dynamic combinations of various control signals. It uses a novel double-cycle controller for initial condition ranking, integrates a Multimodal Large Language Model (MLLM) to optimize condition ordering, and utilizes a flexible multi-control adapter to modulate control signals effectively. This results in more harmonious and accurate image generation across diverse conditions.

**Strengths:**

1. The paper clearly identifies limitations of existing controllable generation methods and introduces a dynamic condition selection mechanism.
2. DynamicControl outperforms existing methods in terms of controllability and image-text alignment by dynamically selecting relevant control conditions. By using MLLMs and a multi-control adapter, it adapts to various control signals, improving versatility in generating images under different conditions.
3. The double-cycle controller provides interpretable and measurable consistency supervision.
4. Extensive experiments and visualizations demonstrate that as high-scoring conditions are progressively added, image quality and text–image alignment improve notably. The ablation study further confirms that the MLLM evaluator significantly enhances controllability.
5. The method generalizes well to unseen control types at inference time, adaptively re-ranking and activating compatible controls for high-quality generation.

**Weaknesses:**

In practical scenarios, it is rare for the same image to require multiple control conditions, such as depth maps, normal maps, or segmentation maps. Most real-world applications in text-to-image generation often use simpler control signals, and the added complexity of handling various condition combinations may not always provide significant benefits for practical use cases.

**Questions:**

1. How does it deal with situations where control signals are **highly contradictory** (e.g., conflicting textures or layouts)?

2. Can this method handle complementary (not conflicting) conditions? For example, depth outlines the whole background while edges draw some foreground objects.

---

### Official Review · Reviewer_xc57 · 2025-10-31

**Soundness:** 3
**Presentation:** 2
**Contribution:** 2
**Rating:** 2
**Confidence:** 5

**Summary:**

The paper presents DynamicControl, a framework that enhances the controllability of text-to-image diffusion models by dynamically selecting and integrating multiple control conditions. Unlike prior methods that use fixed or inefficient control schemes, DynamicControl introduces a double-cycle controller for initial condition evaluation, a Multimodal Large Language Model (MLLM) for ranking, and a multi-control adapter to adaptively fuse diverse visual signals. Experiments show that the method achieves strong controllability, text alignment, and image quality compared to existing approaches.

**Strengths:**

**Promising Performance:** DynamicControl demonstrates strong effectiveness through comprehensive experiments using metrics such as FID, CLIP Score, and condition-specific measures (e.g., SSIM, mIoU). Results across multiple datasets and tasks show clear improvements in controllability and image quality.

**Weaknesses:**

**Clarity of Presentation:** The methodology sections (3.1–3.3) could be clearer, especially in explaining the condition evaluator and multi-control adapter. Some symbols in the equations are undefined, which may hinder understanding. Also, Table A4 has wrong captions.

**Unclear Effectiveness of Multi-Control Adapter:** The ablation results (Table A4) show almost no performance difference between the base model and versions without the MoE or criss-cross attention modules. This suggests that the proposed multi-control adapter contributes minimally to the overall improvement, raising concerns about the clarity and effectiveness of this component’s design and impact. In this situation, it is recommended to remove the *Flexible Multi-Control Adapter* from the core contribution.

**Questionable Novelty Claim**: The paper highlights “new insight” as a main contribution, but this claim is debatable. The core advantage of the work lies in using an MLLM-based condition filtering mechanism to improve generation fidelity under multiple conditions—a concept that aligns closely with existing designs for unified models such as Bagel[1], MetaQuery[2], BLIP3-O[3], and OmniGen[4]. Thus, the proposed approach appears more as an incremental adaptation of established MLLM-guided control strategies rather than a fundamentally new insight.

Reference:
1. Deng et al., Emerging Properties in Unified Multimodal Pretraining, Link: https://arxiv.org/abs/2505.14683
2. Pan et al., Transfer between Modalities with MetaQueries, Link: https://arxiv.org/pdf/2504.06256
3. Chen at al., BLIP3-o: A Family of Fully Open Unified Multimodal Models—Architecture, Training and Dataset, Link: https://arxiv.org/pdf/2505.09568
4. Xiao et al., OmniGen: Unified Image Generation, Link: https://arxiv.org/pdf/2409.11340

**Questions:**

See weakness.

---

### Note · Authors · 2025-11-12

I have read and agree with the venue's withdrawal policy on behalf of myself and my co-authors.